# Differences in Antiretroviral Adherence Behaviors, Treatment Success, and Eligibility for Long-Acting Injectable Treatment between Patients Who Acquired HIV in Childhood vs. Those Who Acquired It in Adolescence/Early Adulthood

**DOI:** 10.3390/jpm12091390

**Published:** 2022-08-27

**Authors:** Anthony Marcellin, Valérie Martel-Laferrière, Anne-Geneviève Genest, Bertrand Lebouché, Suzanne Marcotte

**Affiliations:** 1Faculty of Pharmacy, Aix-Marseille University, 13284 Marseille, France; 2Division of Microbiology and Infectious Diseases and Centre de Recherche du Centre Hospitalier de l’Université de Montréal (CRCHUM), Montréal, QC H2X0C1, Canada; 3Centre Intégré de Santé et de Services Sociaux de la Montérégie-Centre, Greenfield Park, QC J4V2H1, Canada; 4McGill University Health Centre (MUHC), Montréal, QC H4A3J1, Canada; 5Department of Pharmacy, Centre Hospitalier de l’Université de Montréal (CHUM), Montréal, QC H2X0C1, Canada

**Keywords:** HIV, compliance, viral resistances, long-acting injectable antiretrovirals, youth

## Abstract

This study investigates the impact of the age at which HIV was acquired on adherence. There was no difference in adherence between patients who acquired HIV in childhood vs. those who acquired it in adolescence/early adulthood (83% vs. 90%; *p* = 0.24), but achievement of virological/immunological efficacy (78.8% vs. 93.5%, *p* = 0.02) was less likely in patients who had acquired HIV in childhood. On the basis of resistance, patients who acquired HIV in adolescence/early adulthood tended to be more eligible for cabotegravir/rilpivirine treatment (90.3% vs. 80.3%; *p* = 0.11).

## 1. Introduction

Despite simpler and more tolerable regimens for treating HIV infection, suboptimal adherence remains common, particularly among young patients [1,2,3,4,5,6,7]. Young adults may face multiple challenges, such as identity issues, unveiling, lack of a family model, and thinking in terms of short-term benefits [8,9]. This study aims to compare adherence depending on the timing of infection acquisition, i.e., infancy/childhood versus adolescence/early adulthood, to explore the causes of non-adherence and to compare the two groups in terms of the HIV viral load, resistance to antiretroviral therapies, and eligibility for injectable cabotegravir–rilpivirine.

## 2. Materials and Methods

This study included a retrospective chart review and a survey in order to identify factors of non-adherence from the point of view of the healthcare team, and that of the patient, respectively.

### 2.1. Retrospective Study

This study included all patients infected with HIV younger than 25 years of age, currently aged between 18 and 30 (as of 1 May 2021), and seen in three Canadian clinics in the Montreal area (Centre Hospitalier de l’Université de Montréal (CHUM), McGill University Health Center (MUHC), and Charles LeMoyne Hospital (CLMH)). Patients for whom no antiretroviral had been prescribed, or lacking minimal biological data in the medical file (HIV viral load and CD4 count) were excluded. The population was divided into two groups: patients with HIV diagnosed before the age of 10 (group 1), and patients diagnosed with HIV between the ages of 10 and 25 (group 2). Data were collected between 1 June 2021 and 27 August 2021, in Montreal. For the cross-sectional study, patients already receiving long-acting injectable antiretrovirals were excluded.

Data collected included adherence to antiretrovirals as systematically reported in the health care providers’ notes, potential viral resistance determined by genotype testing, and the complexity of the antiretroviral regimen. Poor/insufficient adherence was defined as omitting to take 20% or more of monthly doses, based on the minimum threshold of 80% required for therapeutic adherence in the context of antiretrovirals [6]. We defined good immunological and virological efficacy when the viral load was below 200 copies per ml or undetectable, and when the CD4+ T-lymphocytes count was above 200 cells per mm^3^. Eligibility for cabotegravir/rilpivirine was determined on the basis of the absence of specific mutations, i.e., E138A/G/K/Q/R, K101P/E/Q, V179L, H221Y, F227C, M230I/L, and Y181C/I/V for rilpivirine and S147G, R263K, H51Y, and S153F/Y for cabotegravir, as defined by the Stanford University HIV Drug Resistance Database, and International Antiviral Society-USA [10,11]. In the absence of genotype, patients were assumed to be free of mutations of interest.

### 2.2. Survey

For the cross-sectional study, all patients included in the retrospective chart review were invited, by email or letter, to complete an electronic questionnaire in order to identify self-reported causes of non-adherence to antiretrovirals, except for those already on injectable treatment. With the author’s permission, we used the Godin questionnaire, in English and French, which is validated for antiretroviral therapy (ART) adherence in the Canadian population [12]. It is a brief and simple self-reporting questionnaire evaluating the number of antiretroviral pills missed in the previous 7 days, social activities in the past week (leisure activities, going to bars, restaurants, etc.), and impact on ART adherence (Appendix A). In addition, we included a visual analog scale and some questions to explore the causes of non-adherence and the patient’s interest in a long-acting injectable treatment. The RedCap^®^ software (survey function) was used to collect the data. An HTML link to the questionnaire and a QR-Code with the same function were included in the letter or email inviting participation in the study. To encourage participation, for those who used email for communication, the email was sent 3 times at 2-week intervals. For the others, a letter inviting participation was sent twice by postal mail. About half of the patients were also invited by phone to complete the questionnaire, as this was permitted by the ethics committee at only one of the three participating sites. Informed consent was signed electronically by answering the first question. Participants had the opportunity to enter a draw for a CAD 100 gift card. This study received multicentric approval by the CHUM institutional review and research ethics board.

### 2.3. Statistical Analysis

Groups were described using means, with a 95% confidence interval (95% CI), and numbers, with proportions, for continuous and categorical variables, respectively. The Mann–Whitney U test was used for a comparative analysis of continuous variables, while chi-square tests and Fisher’s exact test were used for a comparative analysis of categorical variables. Analyses were performed using SPSS Statistics v.26 (IBM Corporation, Armonk, NY, USA).

## 3. Results

Out of 148 patients screened, 128 were included (CHUM = 29, MUHC = 92, and CLMH = 7). The reasons for exclusions were: diagnosis of HIV infection over the age of 25 (*n* = 14), inability to determine the date of diagnosis (*n* = 4), insufficient biological data in the medical file (*n* = 1), and absence of a prescribed antiretroviral (*n* = 1). In all, 66 patients were included in group 1 (HIV diagnosed before the age of 10) and 62 in group 2 (patients diagnosed between the ages of 10 and 25) (Table 1).

### 3.1. Retrospective Study

HIV transmission was mainly perinatal in group 1 (95.5%) and sexual in group 2 (88.7%). No transmission through injection drug use was noted in the charts, but 10 patients had acquired HIV by another or unknown route.

The two most commonly used regimens were tenofovir alafenamide/emtricitabine/bictegravir (44 patients; 34%) and lamivudine/abacavir/dolutegravir (34 patients; 27%). Adherence to treatment did not statistically differ between the groups (group 1: 83% vs. group 2: 90%; *p* = 0.24), but there were significantly fewer patients with good immunological and virological ART efficacy in group 1 compared to patients in group 2 (78.8% vs. 93.5%; *p* = 0.02). Most patients in group 2 (93.5%) reached virological suppression against only 81.8% in the group 1 (*p* = 0.05). Overall, independently of their group, patients with poor or insufficient adherence were more at risk of poor or incomplete immuno-virological efficacy (76.5% vs. 4.5%; *p* < 0.001).

In terms of viral mutations, we observed significantly higher drug resistance in group 1 (mean number of pharmacological classes impacted by resistance mutations: 1.17 (95% CI: 0.81–1.52) vs. 0.38 (95% CI: 0.19–0.56); *p* = 0.002). Nucleotide/nucleoside reverse transcriptase inhibitors (NRTIs) resistance (47.9% vs. 2.5%; *p* = 0.001) were more likely in group 1 compared to group 2.

Regarding the eligibility of patients for long-acting injectable cabotegravir/rilpivirine on the basis of resistance, there seemed to be more eligible patients in group 2, but this relation was not statistically significant (80.3% vs. 90.3%; *p* = 0.11). Nine patients (six in group 1 and three in group 2) had mutations that made them ineligible for treatment with cabotegravir/rilpivirine: E138A/K (*n* = 4), K101P/E/Q (*n* = 4), and Y181C (*n* = 1) (Table 2).

### 3.2. Cross-Sectional Study

Four patients were excluded from the cross-sectional study because they were already receiving a long-acting injectable antiretroviral combination. In group 1, out of the 65 who received the questionnaire, 16 people responded (25%). In group 2, out of the 59 who received the questionnaire, 15 people responded (25%).

We did not find a statistically significant association between group and treatment adherence considering the mean number of missed pills over 2 days (0.25 (95% CI: 0–0.61) vs. 0.11 (0–0.33); *p* = 0.95), and the mean number of missed pills over 7 days (0.56 (95% CI: 0–1.18) vs. 0.20 (0–0.43); *p* = 0.57). There was, however, a significant difference between adherence noted in their files and self-reported adherence, since only 45% of the patients reported the same level of adherence as that assessed by a health professional (prescribing physician, clinical pharmacist, or clinical nurse).

Among patients with “optimal” chart adherence, 72% self-rated their adherence as such. Noticeably, all patients whose adherence was judged to be “poor or insufficient” considered their adherence to be “optimal” and reported no omissions over 7 days. Eleven patients considered their adherence to be optimal, although it was described as imperfect or insufficient by health professionals. Conversely, five patients considered that their treatment compliance was not optimal, while the healthcare professional considered it as optimal.

The patients included in group 1 generally reported more social activities, and among them, two said that these situations interfered with taking their antiretroviral treatment properly. Compared to three patients in group 2, five patients in group 1 reported that ART negatively impacted their quality of life. In both groups, the main self-reported cause of non-adherence was simple forgetfulness (59%). Patients also mentioned interference with social life (12%), lack of time to take the medication (10%), and the fear of a possible disclosure of their HIV-positive status (10%) as being etiologies of non-adherence. The semi-structured questionnaire identified additional causes of non-adherence as forgetting renewal at the pharmacy (*n* = 1), going outside the home without bringing along the pills (4%), and “Taking a daily treatment for a pathology imposed since birth” (mentioned by one patient). Another patient reported the size of the pills as an obstacle to adherence.

Regarding interest in long-acting ART, patients in group 2 were more likely (73.3%) to be interested in the injectable cabotegravir/rilpivirine combination than patients in group 1 (56.3%), but this difference was not significant (*p* = 0.32) (Table 3).

## 4. Discussion

Our study has found an association between adherence and immune-virological efficacy. The relation between adherence and virological efficacy is well established in the literature [4,6,13]. Our data do not support the assumption that adherence is different depending on the age of seroconversion. Similarly, Vanthournout et al. [4] did not find a significant association between ART adherence, history of treatment failure, the clinical stage of the pathology, age at diagnosis disclosure, and the duration of taking ART. However, fewer patients with perinatally acquired HIV achieved immunological and virological controls, and their virus showed more resistance mutations, as expected considering the difference between the presumed mean duration of infection in the two groups. Consequently, ART regimens were significantly more complex in this group.

Clinicians see the new injectable therapies as an attractive option for improving adherence, particularly among the younger population, who often do not take other medications. Indeed, 65% of those surveyed expressed interest. However, compared with those who had acquired HIV later, there was a trend toward lower eligibility among those with perinatally acquired HIV, as their virus had more accumulated mutations. This is not surprising, as they had both longer exposure to antiretrovirals and exposure to older agents.

Adolescents and young adults living with HIV are exposed to periods of great change, particularly through puberty. Thus, it is necessary to consider the barriers to adherence related to developmental factors: physical, cognitive, social, emotional changes, combined with an emerging recognition of their sexual identity [8]. These barriers to adherence include in particular motivational barriers related to the acceptance of the disease, and factors related to the impact of potential stigmatization [8]. Social barriers to compliance are favored by chaotic and unstructured lifestyles (disorganization, drug addiction, etc.), material constraints, or even a lack of support from the patient’s social environment (family and friends) [9]. In our study, more than half the respondents reported a negative impact of their antiretrovirals on their quality of life.

This study is not without limitations, especially with regard to the survey component. The 25% response rate to the questionnaire may have affected representativeness. Our study may have a response bias. For example, people who are more involved in their healthcare or with a less busy schedule may be more likely to respond. Potential participants were contacted multiple times, and, in our opinion, more solicitation would not have been acceptable. We do not believe that a written questionnaire would have improved the response rate, as our study population was under 30 years of age. Moreover, it would have posed greater confidentiality issues. In addition, in both groups, on average, higher ART adherence was noted when it was self-assessed using the online questionnaire, compared to adherence according to the notes in the medical file, probably showing a social desirability bias. Finally, due to the small sample, our study might have been underpowered to detect small differences between groups.

In conclusion, we did not find a difference in antiretroviral adherence between patients who acquired HIV in childhood vs. those who acquired it in adolescence/early adulthood. Regardless of the transmission mode, young people living with HIV need appropriate support and methods to reduce various adherence barriers, and to maximize their chances of obtaining a long-term undetectable viral load. Even though many young patients may be interested in long-acting injectable antivirals, it is important to look at previous genotypes to ensure eligibility.

## Figures and Tables

**Table 1 jpm-12-01390-t001:** Demographics.

	Dx before Age 10 (*n* = 66)		Dx between 10 and 25 (*n* = 62)	
	Years	SD	Years	SD
Age	23.97	3.13	26.50	2.24
Range	17–29	/	21–29	/
Presumed duration of infection	22.60	5.18	5.12	3.31
	* **n** *	**%**	* **n** *	**%**
*Gender*				
Female	35	53%	16	26%
Male	30	45%	44	71%
Genderfluid, transgender, non-binary	1	2%	2	3%
*Transmission mode*				
Ante- or perinatal	63	95.5%	0	0.0%
Sexual transmission	0	0.0%	55	88.7%
Injection drug use (IDU)	0	0.0%	0	0.0%
Other, unknown	3	4.5%	7	11.3%
*Regimen complexity*				
One pill, once a day (Single-Tablet Regimens)	44	66.7%	58	93.5%
More than one pill, once a day	18	27.3%	1	1.6%
Other regimen	4	6.1%	3	4.8%

Dx, Diagnosed; SD, Standard-Deviation.

**Table 2 jpm-12-01390-t002:** Comparative data for the retrospective study.

	Dx before Age 10 (*n* = 66)		Dx between 10 and 25 (*n* = 62)	
Retrospective Section	N	%	N	%
*Treatment adherence (according to the medical file)*				
Optimal	25	37.9%	26	41.9%
Good	30	45.5%	30	48.4%
Poor, inadequate	11	16.7%	6	9.7%
*Immunovirological efficacy*				
Good immunological and virological efficacy	52	78.8%	58	93.5%
Good immunological but poor virological efficacy	6	9.1%	4	6.5%
Good virological but poor immunological efficacy	2	3.0%	0	0.0%
Poor immunological and virological efficacy	6	9.1%	0	0.0%
*Resistance in each class of antiretroviral*				
NRTI resistance ^1^	23/48	47.9%	1/40	2.5%
NNRTI resistance (excluding RPV) ^1^	19/48	39.6%	11/40	27.5%
PI resistance ^2^	10/48	20.8%	2/40	5.0%
INSTI resistance ^3^	4/21	19.0%	1/20	5.0%
RPV resistance ^4^	10/53	18.9%	6/40	15.0%
*Eligibility for CAB/RPV LA*				
Eligibility for long-acting antiretroviral therapy	53	80.3%	56	90.3%

Dx: Diagnosed; NRTI: Nucleoside Reverse Transcriptase Inhibitor; NNRTI: Non-Nucleoside Reverse Transcriptase Inhibitor; PI: Protease Inhibitor; INSTI: Integrase Strand Transfer Inhibitor; RPV: Rilpivirine. ^1^ Reverse transcriptase resistance mutations (excluding RPV) M184V (23:0), K103T/N/S/R (14:4), T215Y/D/L/C/F (12:0), M41L (8:0), V179I/D/E/T (6:3), K219E/R/Q (5:0), K70/E/T/R (4:1), T69N/D/A (4:0), L74I (4:0), V106I/M (3:3), P225H (3:1), N348I (2:2), F227L (2:1), G190A (2:1), V75I/T (2:0). ^2^ PI resistance mutations M36I/L (24:3), L63A/P/S/M/V (16:4), K20I/R/T/M (13:4), I13V (11:4), L89M (10:3), L10I/F/V/R (10:3), V82I/L/A/F (8:0), I54M/L/V (6:1), I93L (6:1), L33F/V (6:0), L90M (5:1), I47V (3:0), V32I (2:0), Q58E (2:0). ^3^ INSTI resistance mutations N155H (2:0), V151I (1:0), L74I (0:2), M50I (0:1). ^4^ RPV resistance mutations E138A/K (2:2), K101P/E/Q (3:1), Y181C (1:0). *(x:y) x is the number of patients carrying this mutation in group 1: y is the number of patients carrying this mutation in group 2*.

**Table 3 jpm-12-01390-t003:** Comparative data for the cross-sectional study.

	Dx before Age 10 (*n* = 16)		Dx between 10 and 25 (*n* = 15)	
Cross-sectional Study	Average	95%CI	Average	95% CI
Number of missed pills over 2 days	0.25	0–0.61	0.13	0–0.33
Number of missed pills over 7 days	0.56	0–1.18	0.20	0–0.43
Self-rated importance of antiretroviral treatment	9.67	9.17–10.16	9.53	8.95–10.12
	**N**	**%**	**N**	**%**
Perception of treatment as essential for health	15	93.8%	14	93.3%
Interference between social activities and adherence to antiretrovirals	2	12.5%	0	0.0%
Negative impact on quality of life	5	31.3%	3	20.0%
Interest in long-acting injectable antiretroviral treatment	9	56.3%	11	73.3%

Dx, Diagnosed; 95% CI, 95% Confidence Interval.

## Data Availability

Not applicable.

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
