# Peer review of "Differences in Antiretroviral Adherence Behaviors, Treatment Success, and Eligibility for Long-Acting Injectable Treatment between Patients Who Acquired HIV in Childhood vs. Those Who Acquired It in Adolescence/Early Adulthood"

_jpm, 2022, doi:10.3390/jpm12091390_

Round 1

Reviewer 1 Report

First, I want to congratulate the authors for addressing such an important issue in young adults living with HIV.  

On this paper, I noticed that two different types of outcomes were measured: adherence and eligibility to injectable therapy. I am not sure if I understand well the rationale for combining these outcomes in the same paper. I would recommend to consider splitting them, which will make it easier for the reader to understand the subtleties of the conclusions on adherence. 

In general, I propose to add more information on the survey methods. A validated adherence measure tool was used (Godin) which was bonified by unvalidated other outcome measures. It introduces a potential internal bias. At a minimum, it should be discussed. The questionnaire should be added as an appendix. In addition, more information should be provided to address how the population was identified and reached, to describe measures to maximize response rate, to quantify completion rate in addition to response rate, etc.

A response rate of 25% is low, regardless of the sensitivity of the topic.  Response rate of electronic surveys is lower than paper. Authors should address the potential for nonresponse bias. (ref: Fincham Am J Pharm Educ 2008;72(2):43)

With regards to objective associated with the eligibility to CAB/RPV, authors should justify the selection of RPV RAMs, and why the IAS RAMs (K101E/P, E138A/G/K/Q/R, V179L, Y181C/I/V, H221Y, F227C, and M230I/L)  and used in other publications to predict CAB/RPV failure (AIDS. 2021 Jul 15; 35(9): 1333–1342.).

Reviewer 2 Report

The article is interesting, simple, and straightforward.

Some corrections should be made to clarify aspects of the methodology and facilitate reading the article.

In the survey section, it is necessary to indicate how many people were offered participation and how many of them answered. 

In line 70, when a reference is made to groups 1 and 2, what it refers to into parentheses to facilitate the article's reading.

In the material and methods, you should explain the statistical analysis that has been carried out. There are probabilities and adjusted ORs in the results, but nowhere is it indicated how they had been calculated.

In the results section, there is a probability, but the performed test is not mentioned. It is necessary to identify it. There are also adjusted Odds ratios. It is required to indicate the variables by which they have been adjusted, for example, age and sex.

In line 95 (0.25 ± 0.66), it is not known if the +/- 0.66 is the standard deviation or the 95% confidence interval. This needs to be clarified.

Round 2

Reviewer 1 Report

Methods: Authors selected to use 2 methods (questionnaires and retrospective chart review) to answer the research questions (not clearly described).  As per the last sentence in the introduction they are

- compare adherence depending on the timing of infection acquisition, i.e., infancy/childhood versus adolescence/early adulthood,

- to explore the causes of non-adherence and to compare the two groups regarding HIV viral load, resistance to antiretroviral therapies, and eligibility for injectable cabotegravir–rilpivirine.

ISSUES: There are a lot of analyses comparing 2 groups of young patients with regard to the time of HIV acquisition. This is not reflected in the title.

Population:  According to the inclusion criteria, patients may be younger than 18yo (ie include adolescent/child/infant). What was the age of the participants in each group?  The title specifies young adults, it would need a correction.

Also, a very important determinant is the calendar time of HIV acquisition. Treatment differed very significantly between 1991 and 2010. This will have an impact on RAMs, and virologic and immunologic outcomes. All subsequent analyses may be biased if this factor is not carefully considered.  

Specific to each part of the study:

In the retrospective review, use of STR, adherence between groups and virologic/immunologic response, VL suppression was assessed. A new analysis of poor adherence (defined as >20% non adherence) and poor or imcomplete efficacy (not defined). Mutations are described and compared between groups. finally eligibility to CAB/RPV based on RAMs is compared. 

1- How was adherence assessed in the retrospective review? HCP self-report, pill count, refill history, patient self-report? This is a key component of the study. It needs to be rigourous. Another important pitfall is the period of time during which the adherence was assessed. It was done over a period of 3 months. It is a snapshot assessment of adherence and clinical outcome.  

2- Adherence: did not differ between groups. An association was found between IMM/VIROL response and adherence. This may have been driven by immunologic response, for which CD4 nadir is an important cofactor.  the sentence at lines 246-247 is unclear.  

3- use of STR: it is compared but I do not see the pertinence. If it is to support adherence differences, it should be discussed. If it is just to describe your population, then it should be included in the baseline characteristic table and not compared with statistics.

4- mutations: should be described in table 2

Questionnaire data

Adherence: no differences noted between groups and validated measure of missed pills over 2 days or 7 days. But a discordance between the survey and what is reported in the chart. 

Descriptive factors/behaviors: informative and original. no particular comments. 

Patient interests to CAB/RPV; interesting but underpowered. 

Discussion:

In general, this section should be expanded. Only 2 paragraphs on adherence and VL/CD4 correlation, and on adherence between 2 groups. What about behaviors/barriors? Also discussion should include the above comments/limitations. 

Adherence: both the questionnaire and retrospective review failed to show a difference between groups. Your data DID NOT show a relationship with adherence and viral load (p=0.05). It did with the surrogate of VL and CD4 count. 

line 624-627: is this statement about this study and Vanthournout?

The conclusion should match better the objectives and results. 

Tables

There are no baseline characteristics table (table1). It should be separate from the results. Table 2 should focus on results. I would suggests to make 2 tables for results (1 for the retrospective and 1 for the questionnaire)

Author Response

We would like to thank you for your very pertinent comments which we believe have improved the clarity of this document. We hope that this document meets your expectations.
